# WaveQ: Gradient-Based Deep Quantization of Neural Networks through Sinusoidal Regularization

## Abstract

Deep quantization of neural networks below eight bits can lead to superlinear benefits in storage and compute efficiency. However, homogeneously quantizing all the layers to the same level does not account for the distinction of the layers and their individual properties. Heterogeneous assignment of bitwidths to the layers is attractive but opens an exponentially large non-contiguous hyperparameter space (Available Bitwidths$^{\text{\# Layers}}$). Thus, finding the bitwidth while also quantizing the network to those levels becomes a major challenge. This paper addresses this challenge through a sinusoidal regularization mechanism, dubbed WaveQ. Adding our parametrized sinusoidal regularizer enables WaveQ to not only find the quantized weights, but also learn the bitwidth of the layers by making the period of the sinusoidal regularizer a trainable parameter. In addition, the sinusoidal regularizer itself is designed to align its minima on the quantization levels. With these two innovations, during training, stochastic gradient descent uses the form of the sinusoidal regularizer and its minima to push the weights to the quantization levels while it is also learning the period which will determine the bitwidth of each layer separately. As such WaveQ is a gradient-based mechanism that jointly learns the quantized weights as well as the heterogeneous bitwidths. We show that WaveQ balances compute efficiency and accuracy, and provides a heterogeneous bitwidth assignment for quantization of a large variety of deep networks (AlexNet, CIFAR-10, MobileNet, ResNet-18, ResNet-20, SVHN, and VGG-11) that virtually preserves the accuracy. WaveQ is versatile and can also be used with predetermined bitwidths by fixing the period of the sinusoidal regularizer. In this case, WaveQ, on average, improves the accuracy of quantized training algorithms (DoReFa and WRPN) by $\sim 4.8\%$, and outperforms multiple state-of-the-art techniques. Finally, WaveQ is applicable to quantizing transformers and yields significant benefits.

## 1 Introduction

Quantization, in general, and deep quantization (below eight bits) (Krishnamoorthi, 2018), in particular, aims to not only reduce the compute requirements of DNNs but also reduce their memory footprint (Zhou et al., 2016; Judd et al., 2016b; Hubara et al., 2017; Mishra et al., 2018; Sharma et al., 2018). Nevertheless, without specialized training algorithms, quantization can diminish the accuracy. As such, the practical utility of quantization hinges upon addressing two fundamental challenges: (1) discovering the appropriate bitwidth of quantization for each layer while considering the accuracy; and (2) learning weights in the quantized domain for a given set of bitwidths.

This paper formulates both of these challenges as a *gradient-based* joint optimization problem by introducing an additional novel sinusoidal regularization term in the training loss, called WaveQ. The following two main insights drive this work. (1) Sinusoidal functions ($sin^2$) have inherent periodic minima and by adjusting the period, the minima can be positioned on quantization levels corresponding to a bitwidth at per-layer granularity. (2) As such, sinusoidal period becomes a direct and continuous representation of the bitwidth. Therefore, WaveQ incorporates this continuous variable (i.e., period) as a differentiable part of the training loss in the form of a regularizer. Hence, WaveQ is a differentiable regularization mechanism, it piggy backs on the stochastic gradient descent that trains the neural network to also learn the bitwidth (the period). Simultaneously this parametric sinusoidal regularizer pushes the weights to the quantization levels ($sin^2$ minima).

By adding our parametric sinusoidal regularizer to the original training objective function, our method automatically yields the bitwidths for each layer along with nearly quantized weights for

those bitwidths. In fact, the original optimization procedure itself is harnessed for this purpose, which is enabled by the differentiability of the sinusoidal regularization term. As such, quantized training algorithms (Zhou et al., 2016; Mishra et al., 2018) that still use some form of backpropagation (Rumelhart et al., 1986) can effectively utilize the proposed mechanism by modifying their loss. Moreover, the proposed technique is flexible as it enables heterogeneous quantization across the layers. The WaveQ regularization can also be applied for training a model from scratch, or for fine-tuning a pretrained model.

In contrast to the prior inspiring works (Uhlich et al., 2019; Esser et al., 2019), WaveQ is the only technique that casts finding the bitwidths and the corresponding quantized weights as a simultaneous gradient-based optimization through sinusoidal regularization during the training process. We also prove a theoretical result to provide an insight on why the proposed approach leads to solutions preserving the original accuracy during quantization. We evaluate WaveQ using different bitwidth assignments across different DNNs (AlexNet, CIFAR-10, MobileNet, ResNet-18, ResNet-20, SVHN, and VGG-11). To show the versatility of WaveQ, it is used with two different quantized training algorithms, DoReFa (Zhou et al., 2016) and WRPN (Mishra et al., 2018). Over all the bitwidth assignments, the proposed regularization, on average, improves the top-1 accuracy of DoReFa by 4.8%. The reduction in the bitwidth, on average, leads to 77.5% reduction in the energy consumed during the execution of these networks. Finally, we apply WaveQ to Transformer DNNs citeppby augmenting their loss with WaveQ parametric sinusiodal regularization. In this case, the conventional stochastic gradient descent plus WaveQ regularization is used to quantize the big Transformer model from (Ott et al., 2018) for machine translation on the IWSLT14 German-English dataset (IWS). For 5, 6, and 7-bit quantization, training with WaveQ yields 0.46, 0.14, 0.04 improved BiLingual Evaluation Understudy (BLEU) score, respectively. As a point of reference, the original big Transformer model from (Ott et al., 2018) improves the BLEU by only 0.1 over the state-of-the-art. Code available at `https://github.com/waveq-reg/waveq`

## 2 JOINT LEARNING OF LAYER BITWIDTHS AND QUANTIZED PARAMETERS

Our proposed method WaveQ exploits weight regularization in order to automatically quantize a neural network while training. To that end, Section 2.1 describes the role of regularization in neural networks and then Section 2.2 explains WaveQ in more details.

### 2.1 PRELIMINARIES

**Quantizer.** We discuss how quantization of weight works. Consider a floating-point variable $w_f$ to be mapped into a quantized domain using $(b + 1)$ bits. Let $\mathcal{Q}$ be a set of $(2k + 1)$ quantized values, where $k = 2^b - 1$. Considering linear quantization, $\mathcal{Q}$ can be represented as $\left\{-1, -\frac{k-1}{k}, ..., -\frac{1}{k}, 0, \frac{1}{k}, ..., \frac{k-1}{k}, 1\right\}$, where $\frac{1}{k}$ is the size of the quantization bin. Now, $w_f$ can be mapped to the $b$-bit quantization Zhou et al. (2016) space as follows.

$$w_{qo} = 2 \times \text{quantize}_b \left( \frac{\tanh(w_f)}{2 \max(|\tanh(W_f)|)} + \frac{1}{2} \right) - 1 \tag{2.1}$$

In Equation 2.1, $\text{quantize}_b(x) = \frac{1}{2^b-1}\text{round}((2^b - 1)x)$, $w_f$ is a scalar, $W_f$ is a vector, and $w_{qo}$ is a scalar in the range $[-1, 1]$. Then, a scaling factor $c$ is determined per layer to map the final quantized weight $w_q$ into the range $[-c, +c]$. As such, $w_q$ takes the form $cw_{qo}$, where $c > 0$, and $w_{qo} \in \mathcal{Q}$.

**Soft constraints through regularization and the loss landscape of neural networks.** Neural networks' loss landscapes are known to be highly non-convex and it has been empirically verified that loss surfaces for large networks have many local minima that essentially provide equivalent test errors Choromanska et al. (2015); Li et al. (2018). This opens up the possibility of adding soft constrains as extra custom objectives during the training process, in addition to the original objective (i.e., to minimize the accuracy loss). The added constraint could be with the purpose of increasing generalization or imposing some preference on the weights values.

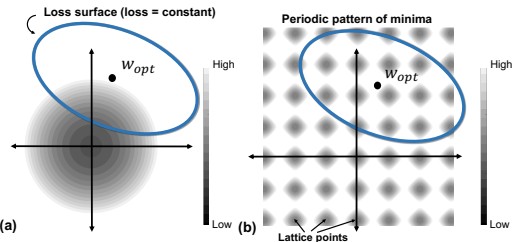

Figure 1: **Sketch for a hypothetical loss surface (original task loss to be minimized) and an extra regularization term in 2-D weight space: for (a) weight decay, and (b) WaveQ.**

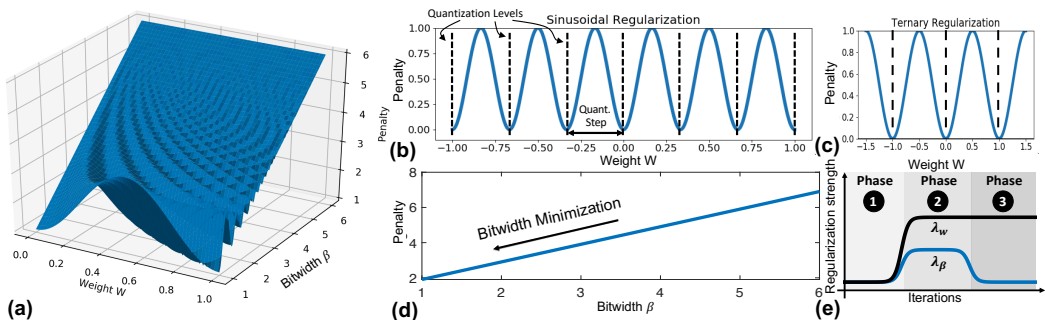

Figure 2: **(a) 3-D visualization of the proposed generalized objective WaveQ. (b) WaveQ 2-D profile,** $w.r.t$ **weights, adapting for arbitrary bitwidths, (c) example of adapting to ternary quantization. (d) WaveQ 2-D profile** $w.r.t$ **bitwidth. (e) Regularization strengths profiles,** $\lambda_w$**, and** $\lambda_\beta$**, across training iterations.**

**Regularization in action.** Regularization effectively constrains weight parameters by adding a corresponding term (regularizer) to the objective loss function. A classical example is Weight Decay Krogh & Hertz (1991) that aims to reduce the network complexity by limiting the growth of the weights. This soft constraint is realized by adding a term, proportional to the $L_2$ norm of the weights to the objective function as the regularizer that penalizes large weight values. WaveQ, on the other hand, uses regularization to push the weights to the quantization levels. For the sake of simplicity and clarity, Figure 1(a) and (b) illustrate a geometrical sketch for a hypothetical loss surface (original objective function to be minimized) and an extra regularization term in 2-D weight space, respectively. For weight decay regularization (Figure 1 (a)), the faded circle shows that as we get closer to the origin, the regularization loss is minimized. The point $w_{opt}$ is the optimum just for the loss function alone and the overall optimum solution is achieved by striking a balance between the original loss term and the regularization loss term. Similarly, Figure 1(b) shows a representation of the proposed periodic regularization for a fixed bitwidth $\beta$. A periodic pattern of minima pockets are seen surrounding the original optimum point. The objective of the optimization problem is to find the best solution that is the closest to one of those minima pockets where weight values are nearly matching the desired quantization levels, hence the name quantization-friendly.

## 2.2 WAVEQ REGULARIZATION

The proposed regularizer is formulated in Equation (2.2) where the first term pushes the weights to the quantization levels and the second *correlated* term aims to reduce the bitwidth of each individual layer heterogeneously.

$$R(w;\beta) = \lambda_w \underbrace{\sum_i \sum_j \frac{\sin^2\left(\pi w_{ij}(2^{\beta_i}-1)\right)}{2^{\beta_i}}}_{\text{Weights quantization regularization}} + \lambda_\beta \underbrace{\sum_i \beta_i}_{\text{Bitwidth regularization}} \quad (2.2)$$

In Equation (2.2), $\lambda_w$ is the weights quantization regularization strength which governs how strongly weight quantization errors are penalized, and $\lambda_\beta$ is the bitwidth regularization strength. The parameter $\beta_i$ is proportional to the quantization bitwidth which is elaborated later in this section. Figure 2 (a) shows a 3-D visualization of our regularizer, $R$. Figure 2 (b), (c) show a 2-D profile w.r.t weights ($w$), while (d) shows a 2-D profile w.r.t the bitwidth ($\beta$).

**Periodic sinusoidal regularization.** As shown in Equation equation 2.2, the first regularization term is based on a periodic function (sinusoidal) that adds a smooth and differentiable term to the original objective, Figure 2(b). The periodic regularizer induces a periodic pattern of minima that correspond to the desired quantization levels. Such correspondence is achieved by matching the period to the quantization step ($1/(2^{\beta_i}-1)$) based on a particular number of bits ($\beta_i$) for a given layer $i$.

**Learning the sinusoidal period.** The parameter $\beta_i$ in (Equation 2.2) controls the period of the sinusoidal regularizer. Thereby $\beta_i$ is directly proportional to the actual quantization bitwidth ($b_i$) of layer $i$ as follows:

$$b_i = \lceil \beta_i \rceil, \quad \text{and} \quad \alpha_i = 2^{b_i}/2^{\beta_i} \quad (2.3)$$

In Equation (B.2) where $\alpha_i \in \mathbb{R}^+$ is a scaling factor. Note that $b_i \in \mathbb{Z}$ is the only discrete parameter, while $\beta_i \in \mathbb{R}^+$ is a continuous real valued variable, and $\lceil * \rceil$. is the ceiling operator. While the first term in Equation equation 2.2 is only responsible for promoting quantized weights, the second term ($\lambda_\beta \sum_i \beta_i$) aims to reduce the number of bits for each layer $i$ individually while the

overall loss is aiming to maximize accuracy. As such, this term is a soft constraint that yields heterogeneous bitwidths for different layers. The main insight here is that $\beta_i$, which also controls the period of the sinusoidal term, is a continuous valued parameter by definition. As such, $\beta_i$ acts as an ideal optimization objective constraint and a proxy to minimize the actual quantization bitwidth $b_i$. Therefore, WaveQ avoids the issues of gradient-based optimization for discrete valued parameters. Furthermore, the benefit of learning the sinusoidal period is two-fold. First, it provides a smooth differentiable objective for finding minimal bitwidths. Second, simultaneously learning the scaling factor ($\alpha_i$) associated with the found bitwidth.

Leveraging the sinusoidal properties, WaveQ learns the following two quantization parameters simultaneously: (1) a per-layer quantization bitwidth ($b_i$) along with (2) a scaling factor ($\alpha_i$) through learning the period of the sinusoidal function. Additionally, by exploiting the periodicity, differentiability, and the local convexity profile in sinusoidal functions, WaveQ automatically propels network weights towards values that are inherently closer to quantization levels according to the jointly learned quantizer's parameters $b_i$, $\alpha_i$ as follows.

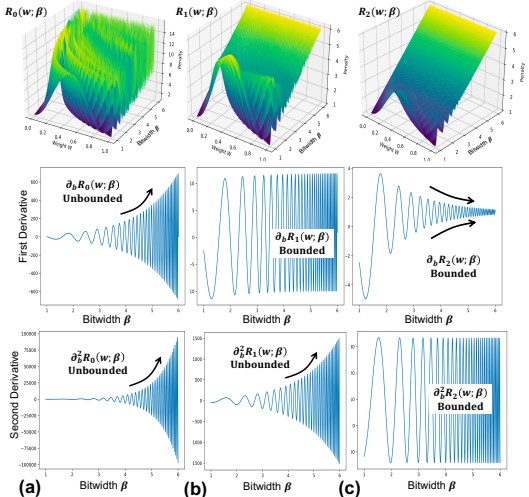

Mapping this to the quantizer's design, for $(b+1)$ bits quantization, $k = 2^b - 1$, and $c = \alpha$. These learned parameters $(b, \alpha)$, as explained above, can be mapped to the quantizer parameters explained in Equation equation 2.1. For $(b+1)$ bits quantization (the extra bit is the sign bit):

$$k = 2^b - 1, \quad \text{and} \quad c = \alpha = 2^b/2^\beta \quad (2.4)$$

**Bounding the gradients.** The denominator in the first term of Equation equation 2.2 ($\sum_i \sum_j \frac{\sin^2\left(\pi w_{ij}(2^{\beta_i}-1)\right)}{2^{\beta_i}}$) is used to control the range derivatives of the proposed regularization term with respect to $\beta$. This denominator is chosen to limit vanishing and exploding gradients during training. To this end, we compared three variants of equation equation 2.2 with different normalization defined, for $k = 0$, 1, and 2, as:

$$R_k(w;\beta) = \lambda_w \sum_i \sum_j \frac{\sin^2\left(\pi w_{ij}(2^{\beta_i}-1)\right)}{2^{k\beta_i}} + \lambda_\beta \sum_i \beta_i$$
$$(2.5)$$

Figure 3: **Visualization for three variants of the proposed regularization objective using different normalizations and their respective first and second derivatives with respect to $\beta$. (a)** $R_0(w;\beta)$, **(b)** $R_1(w;\beta)$, **and (c)** $R_2(w;\beta)$.

Figure 3 (a), (b), (c) provide a visualization on how each of the proposed scaled variants affect the first and second derivatives. For $R_{k=0}$ and $R_{k=2}$, there are regions of vanishing or exploding gradients. Only the regularization $R_{k=1}$ (the proposed one) is free of such issues.

**Setting the regularization strengths.** The convergence behavior depends on the setting of the regularization strengths $\lambda_w$ and $\lambda_\beta$. Our proposed objective seeks to learn multiple quantization parameterization in conjunction. As such, the learning process can be portrayed as three phases (Figure 2(e)). In Phase (❶), the learning process optimizes for the original task loss $E_0$. Initially, the small $\lambda_w$ and $\lambda_\beta$ values allow the gradient descent to explore the optimization surface freely. As the training process moves forward, we transition to phase (❷) where the larger $\lambda_w$ and $\lambda_\beta$ gradually engage both the weights quantization regularization and the bitwidth regularization, respectively. Note that, for this to work, the strength of the weights quantization regularization $\lambda_w$ should be higher than the strength of the bitwidth regularization $\lambda_\beta$ such that a bitwidth per layer could be properly evaluated and eventually learned during this phase. After the bitwidth regularizer converges to a bitwidth for each layer, we transition to phase (❸), where we *fix* the learned bitwidths and gradually decay $\lambda_\beta$ while we keep $\lambda_w$ high. The criterion for choosing $\lambda_w$ and $\lambda_\beta$ is to balance the magnitude of regularization loss to be smaller than the magnitude of accuracy loss. The mathematical formula used to generate $\lambda_w$ and $\lambda_\beta$ profiles can be found in the supplementary material. (Figure 8).

## 3 THEORETICAL ANALYSIS

The results of this section are motivated as follows. Intuitively, we would like to show that the global minima of $E = E_0 + R$ are very close to the minima of $E_0$ that minimizes $R$. In other words, we expect to extract among the original solutions, the ones that are most prone to be quantized. To establish such result, we will not consider the minima of $E = E_0 + R$, but the sequence $S_n$ of minima of $E_n = E_0 + \delta_n R$ defined for any sequence $\delta_n$ of real positive numbers. The next theorem shows that our intuition holds true, at least asymptotically with $n$ provided $\delta_n \to 0$.

**Theorem 1.** *Let $E_0, R : \mathbb{R}^n \to [0, \infty)$ be continuous and assume that the set $S_{E_0}$ of the global minima of $E_0$ is non-empty and compact. As $S_{E_0}$ is compact, we can also define $S_{E_0,R} \subseteq S_{E_0}$ as the set of minima of $E_0$ which minimizes $R$. Let $\delta_n$ be a sequence of real positive numbers, define $E_n = E_0 + \delta_n R$ and the sequence $S_n = S_{E_n}$ of the global minima of $E_n$. Then, the following holds true:*

1. *If $\delta_n \to 0$ and $S_n \to S_*$, then $S_* \subseteq S_{E_0,R}$,*

2. *If $\delta_n \to 0$ then there is a subsequence $\delta_{n_k} \to 0$ and a non-empty set $S_* \subseteq S_{E_0,R}$ so that $S_{n_k} \to S_*$,*

*where the convergence of sets, denoted by $S_n \to S_*$, is defined as the convergence to $0$ of their Haussdorff distance, i.e., $\lim_{n\to\infty} d_H(S_n, S_*) = 0$.*

*Proof.* For the first statement, assume that $S_n \to S_*$. We wish to show that $S_* \subseteq S_{E_0,R}$. Assume that $x_n$ is a sequence of global minima of $F + \delta_n G$ converging to $x_*$. It suffices to show that $x_* \in S_{E_0,R}$. First let us observe that $x_* \in S_{E_0}$. Indeed, let

$$\lambda = \inf_{x \in \mathbb{R}^n} E_0(x)$$

and assume that $x \in S_{E_0}$. Then,

$$\lambda \le E_0(x_n) \le (E_0 + \delta_n R)(x_n) \le (E_0 + \delta_n R)(x) = \underbrace{\lambda + \delta_n R(x)}_{\to \lambda}.$$

Thus, since $E_0$ is continuous and $x_n \to x_*$ we have that $E_0(x_*) = \lambda$ which implies $x_* \in S_{E_0}$. Next, define

$$\mu = \inf_{x \in S_{E_0}} R(x).$$

Let $\hat{x} \in S_{E_0,R}$ so that $R(\hat{x}) = \mu$. Now observe that, by the minimality of $x_n$ we have that

$$\lambda + \delta_n \mu = (E_0 + \delta_n R)(\hat{x}) \ge (E_0 + \delta_n R)(x_n) \ge \lambda + \delta_n R(x_n)$$

Thus, $R(x_n) \le \mu$ for all $n$. Since $R$ is continuous and $x_n \to x_*$ we have that $R(x_*) \le \mu$ which implies that $R(x_*) = \mu$ since $x_* \in S_{E_0}$. Thus, $x_* \in S_{E_0,R}$. The second statement follows from the standard theory of Hausdorff distance on compact metric spaces and the first statement. $\square$

Theorem 1 implies that by decreasing the strength of $R$, one recovers the subset of the original solutions that achives the smallest quantization loss. In practice, we are not interested in global minima, and we should not decrease much the strength of $R$. In our context, Theorem 1 should then be understood as a proof of concept on why the proposed approach leads the expected result. Experiments carried out in the next section will support this claim. Additionally, note that while the theorem is stated in terms of a limit as the regularization parameter vanishes, the proof in fact gives a corresponding stability result. Namely, if the regularization parameter is sufficiently small relative to the main loss then the minimizers will be "almost" quantized. For the interested reader, we provide a more detailed version of the above analysis in the supplementary material.

## 4 EXPERIMENTAL RESULTS

To demonstrate the effectiveness of WaveQ, we evaluated it on several deep neural networks with different Image Classification datasets (CIFAR10, SVHN, and ImageNet), and one Transformer-based network, which is the big Transformer model from(Ott et al., 2018) on the IWSLT14 German-English dataset (IWS). We provide results for two different types of quantization. First, we show quantization results for *learned* heterogeneous bitwidths using WaveQ and we provide different analyses to asses the quality of these learned bitwidth assignments. Second, we further provide results assuming a *preset* homogeneous bitwidth assignment as a special setting of WaveQ. This is, in some cases, a practical assumption that might stem from particular hardware requirements or constraints. Table 1 provides a summary of the evaluated networks and datasets for both learned heterogeneous bitwidths, and the special case of training preset homogeneous bitwidth assignments. We compare our proposed WaveQ method with PACT (Choi et al., 2018a), LQ-Nets (Zhang et al., 2018),

Table 1: **Comparison with state-of-the-art quantization methods on ImageNet. The " W/A " values are the bitwidths of weights/activations.**

| W/A | Benchmark | | AlexNet | | ResNet-18 | | MobileNet-V2 | |
|---|---|---|---|---|---|---|---|---|
| Quantization | Method | Assignment | Top-1 | Top-5 | Top-1 | Top-5 | Top-1 | Top-5 |
| W32/A32 | Full Precision | Homogenous | 57.1 | 80.2 | 70.1 | 89.5 | 71.8 | 90.3 |
| W3/A3 | PACT | Homogenous | 55.6 | - | 68.1 | 88.2 | - | - |
| | LQ-Nets | Homogenous | - | - | 68.2 | 87.9 | - | - |
| | DSQ | Homogenous | - | - | 68.7 | - | - | - |
| | DoReFa | Homogenous | 54.1 | 75.1 | 67.9 | 87.5 | 58.3 | 78.1 |
| W3/A3 | **DoReFa + WaveQ** | **Preset Homogenous** | **55.8** | **77.2** | **68.9** | **89.9** | **60.4** | **83.1** |
| | Improvement | | 0.2%⬆ | 2.1%⬆ | 0.2%⬆ | 1.7%⬆ | 2.1%⬆ | 5.0%⬆ |
| W4/A4 | PACT | Homogenous | 55.7 | - | 69.2 | 89.0 | 61.4 | 83.7 |
| | LQ-Nets | Homogenous | - | - | 69.3 | 88.8 | - | - |
| | DSQ | Homogenous | - | - | 69.6 | - | 64.8 | - |
| | WRPN | Homogenous | 54.9 | 75.4 | 68.8 | 88.1 | 64.3 | 84.5 |
| | DoReFa | Homogenous | 55.5 | 76.3 | 69.1 | 88.5 | 64.6 | 85.1 |
| W4/A4 | **DoReFa + WaveQ** | **Preset Homogenous** | **56.2** | **79.2** | **69.8** | **89.1** | **65.4** | **85.5** |
| | Improvement | | 0.5%⬆ | 2.9%⬆ | 0.2%⬆ | 0.1%⬆ | 0.6%⬆ | 0.4%⬆ |
| W(*Learn*)/A4 | **DoReFa + WaveQ** | **Learned Heterogenous** | **56.5** W3.85 | **79.8** | **70.0** W3.57 | **89.3** | **65.8** W3.95 | **85.8** |
| | Improvement | | 0.3%⬆ | 0.6%⬆ | 0.2%⬆ | 0.2%⬆ | 0.4%⬆ | 0.3%⬆ |
| | **Energy Saving** | | | **2.08x** | | **1.24x** | | **1.78x** |

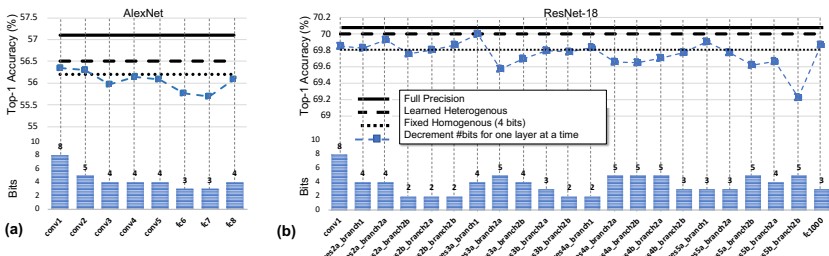

Figure 4: **Quantization bitwidth assignments across layers. (a) AlexNet (average bitwidth = 3.85 bits). (b) ResNet-18 (average bitwidth = 3.57 bits)**

DSQ (Gong et al., 2019), and DoReFa, which are current state-of-the-art methods that show results with homogeneous 3-, and 4-bit weight/activation quantization for various networks (AlexNet, ResNet-18, and MobileNet).

**Experimental setup.** We implemented WaveQ on top of DoReFa inside Distiller (Zmora et al., 2018), an open source framework for quantization by Intel that implements various quantization techniques. The reported accuracies for DoReFa and WRPN are with the built-in implementations in Distiller, which may not exactly match the accuracies reported in their respective papers. However, an independent implementation from a major company provides an unbiased foundation for comparison. We quantize all convolution and fully connected layers, except for the first and last layers which use 8-bits. This assumption is commensurate with the previous techniques.

Table 2: **Accuracies of different networks using plain WRPN, plain DoReFa and DoReFa + WaveQ on homogeneous weight quantization.**

| W/A | Benchmark | SimpleNet on CIFAR10 | ResNet-20 on CIFAR10 | VGG-11 On CIFAR10 | SVHN-8 on SVHN |
|---|---|---|---|---|---|
| Quantization | Method | | Top-1 Accuracy (%) | | |
| W32/A32 | Full Precision | 74.53 | 93.3 | 94.13 | 96.47 |
| W3/A32 | WRPN | 63.44 | 80.28 | 78.56 | 79.36 |
| | DoReFa | 65.13 | 81.57 | 78.78 | 81.45 |
| | **DoReFa + WaveQ** | **73.65** | **92.52** | **93.18** | **95.32** |
| | **Improvement** | 8.52%⬆ | 11%⬆ | 14.4%⬆ | 13.9%⬆ |
| W4/A32 | WRPN | 68.23 | 88.16 | 85.07 | 89.24 |
| | DoReFa | 70.75 | 89.24 | 86.98 | 89.56 |
| | **DoReFa + WaveQ** | **74.14** | **93.01** | **93.96** | **96.12** |
| | **Improvement** | 3.39%⬆ | 3.77%⬆ | 6.98%⬆ | 6.56%⬆ |
| W5/A32 | WRPN | 71.17 | 92.11 | 91.10 | 90.84 |
| | DoReFa | 72.41 | 92.24 | 91.68 | 92.56 |
| | **DoReFa + WaveQ** | **74.45** | **93.13** | **94.11** | **96.42** |
| | **Improvement** | 2.04%⬆ | 0.89%⬆ | 2.43%⬆ | 3.86%⬆ |

## 4.1 LEARNED HETEROGENEOUS BITWIDTHS

As for quantizing both weights and activations, Table 1 shows that incorporating WaveQ into the quantized training process yields best accuracy results outperforming PACT, LQ-Net, DSQ, and DoReFa with significant margins. Furthermore, it can be seen that the learned heterogeneous bitwidths yield better accuracy as compared to the preset 4-bit homogeneous assignments, with

lower, on average, bitwidth (3.85, 3.57, and 3.95 bits for AlexNet, ResNet-18, and MobileNet, respectively). Figure 4(a),(b) (bottom bar graphs) show the learned heterogeneous weight bitwidths over layers for AlexNet and ResNet-18, respectively. As seen, WaveQ parametric regularization yields a spectrum of varying bitwidth assignments to the layers which vary from 2 bits to 8 bits with an irregular pattern. These results demonstrate that the proposed regularization, WaveQ, automatically distinguishes different layers and their varying importance with respect to accuracy while learning their respective bitwidths. To assess the quality of these bitwidths assignments, we conduct a sensitivity analysis on the relatively big networks (see next subsection).

**Benefits of heterogeneous quantization.** Figure 4(a),(b) (top graphs) show various comparisons and sensitivity results for learned heterogeneous bitwidth assignments for bigger networks (AlexNet and ResNet-18). It is infeasible to enumerate these networks' respective quantization spaces. Compared to 4-bit homogeneous quantization, learned heterogeneous assignments achieve better accuracy with lower, on average, bitwidth 3.85 bits for AlexNet and 3.57 bits for ResNet-18. This demonstrates that a homogeneous (uniform) assignment of the bits is not always the desired choice to minimize accuracy loss. Furthermore, Figure 4 also shows that decrementing the learned bitwidth for any single layer at a time results in 0.44% and 0.24% average reduction in accuracy for AlexNet and ResNet-18, respectively. The dotted blue line with ■ markers shows how differently decrementing the bitwidth of various individual layers affect the accuracy. This trend further demonstrates the effectiveness of learning with WaveQ to find the lowest bitwidth that maximizes the accuracy.

**Energy savings.** To demonstrate the energy savings of the solutions found by WaveQ, we evaluate it on Stripes (Judd et al., 2016a), a custom accelerator designed for DNNs, which exploits bit-serial computation to support flexible bitwidths for DNN operations. As shown in Table 1, the reduction in the bitwidth, on average, leads to 77.5% reduction in the energy consumed during the execution of these networks.

## 4.2 PRESET HOMOGENOUS BITWIDTH QUANTIZATION

We also consider a *preset* homogeneous bitwidth quantization which can also be supported by WaveQ under special settings where we fix $\beta$ (to a preset bitwidth). Hence, only the first regularization term is engaged for propelling the weights to the quantization levels.

Table 2 shows accuracies of different networks (SimpleNet-5, ResNet-20, VGG-11, and SVHN-8) using plain WRPN, plain DoReFa and DoReFa + WaveQ for 3, 4, and 5 bitwdiths. As depicted, the results concretely show the effect of incorporating WaveQ into existing quantized training techniques and how it outperforms the previously reported accuracies.

**Weight distributions during training.** Figure 5 shows the evolution of weights distributions over fine-tuning epochs for different layers of CIFAR10, SVHN, AlexNet, and ResNet-18 networks. The high-precision weights form clusters and gradually converge around the quantization centroids as regularization loss is minimized along with the main accuracy loss.

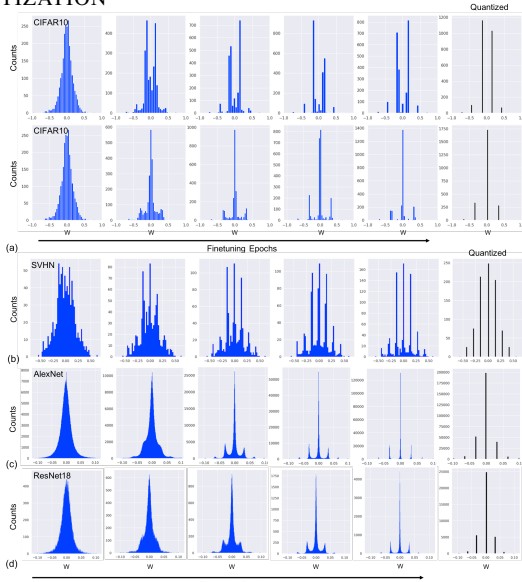

Figure 5: **Evolution of weight distributions over training epochs at different layers and bitwidths for different networks. (a) CIFAR10, (b) SVHN, (c) AlexNet, (d) ResNet18.**

## 4.3 WAVEQ FOR TRANSFORMER QUANTIZATION

Transformers (encoder-decoder architectures) have been shown to achieve best results for NLP tasks including machine translation (Vaswani et al., 2017) and automatic speech recognition (Mohamed et al., 2019). A Transformer layer relies on a key-value self-attention mechanism for learning relationships between distant concepts, rather than relying on recurrent connections and memory cells. Herein, we extend the application of WaveQ regularization to improve the accuracy of deeply

Table 3: **Performance of WaveQ for quantizing Transformers.**

| | BLEU score | |
| --- | --- | --- |
| Bitwidth | Unregularized Training (without WaveQ) | **Regularized Training (with WaveQ)** |
| Full precision | 34.93 | |
| 7-bit | 34.79 | **34.83 (0.04↑)** |
| 6-bit | 34.39 | **34.53 (0.14↑)** |
| 5-bit | 32.74 | **33.20 (0.46↑)** |

quantized (below 8 bits) Transformer models. We run our experiments on IWSLT 2014 German-English (DE-EN) dataset. We use the Transformer model implemented in the `fairseq-py` toolkit (Ott et al., 2019). All experiments are based on the big transformer model with 6 blocks in the encoder and decoder networks. Table 3 shows the effect of applying WaveQ regularization into the training process for 5, 6, and 7-bit quantization on the final accuracy (BLEU score). WaveQ consistently improves the BLEU score of quantized models at various quantization bitwidths (7-5 bits). Moreover, higher improvements are obtained at lower bitwidths.

## 5 DISCUSSION

We conduct an experiment that uses WaveQ for training from scratch. For the sake of clarity, we are considering in this experiment the case of preset bitwidth assignments (i.e., $\lambda_\beta = 0$). Figure 6-Row(I)-Col(I) shows weight trajectories without WaveQ as a point of reference. Row(II)-Col(I) shows the weight trajectories when WaveQ is used with a constant $\lambda_w$.

As the Figure illustrates, using a constant $\lambda_w$ results in the weights being stuck in a region close to their initialization, (i.e., quantization objective dominates the accuracy objective). However, if we dynamically change the $\lambda_w$ following the exponential curve in Figure 6-Row(III)-Col(I)) during the from-scratch training, the weights no longer get stuck. Instead,

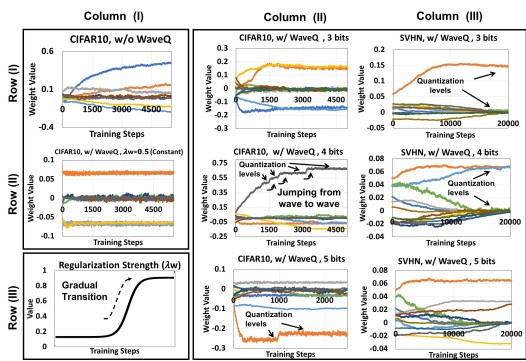

Figure 6: **Weight trajectories. The 10 colored lines in each plot denote the trajectory of 10 different weights.**

the weights traverse the space (*i.e., jump from wave to wave*) as illustrated in Figure 6-Cols(II) and (III) for CIFAR and SVHN, respectively. In these two columns, Rows (I), (II), (III), correspond to quantization with 3, 4, 5 bits, respectively. citepInitially, the smaller $\lambda_w$ values allow the gradient descent to explore the optimization surface freely, as the training process moves forward, the larger $\lambda_w$ gradually engages the sinusoidal regularizer, and eventually pushes the weights close to the quantization levels. Further convergence analysis is provided in the supplementary material.

Next, we provide results comparing two profiles of the regularization strength ($\lambda_w$). *Profile 1*: $\lambda_w$ gradually increases as training proceeds then gradually decays towards the end of training. (Figure 7(a)). *Profile 2*: $\lambda_w$ gradually increases as training proceeds and remains high (Figure 7(c)). Figure 7(a,c) depicts different loss components and Figure 7(b,d) visualizes weights trajectories. Both profiles show that accuracy loss is unimpededly minimized along with WaveQ loss. Our theoretical results align with *Profile 1* (Figure 7(b))–what reviewer insightfully pointed out. Although $\lambda_w$ decays back towards end of training, the weights mostly remain tied to their

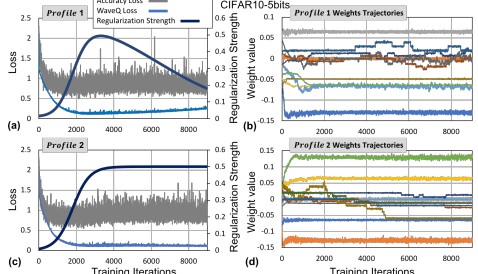

Figure 7: **Weight trajectories and training losses for different $\lambda_w$ profiles.**

quantization levels except for a few deflections that cause slight increase of the regularization loss towards end of training. In terms of test accuracy, both profiles yield similar results (*Profile 1*, 74.95%) vs (*Profile 2*, 74.45%). Note that while the theorem is stated in terms of a limit as the regularization parameter vanishes, the proof in fact gives a corresponding stability result. Namely, if the regularization parameter is sufficiently small relative to the main loss then the minimizers will be "almost" quantized.

## 6 RELATED WORK

This research lies at the intersection of (1) quantized training algorithms and (2) techniques that discover bitwidth for quantization. The following discusses the most related works in both directions. Most related methods define a new optimization problem and use a special method for solving it. For example, (Bai et al., 2019) uses a proximal gradient method (adds a prox step after each stochastic gradient step), (Yang et al., 2020) uses ADMM, and (Tung & Mori, 2018) takes a Bayesian approach. This only makes the training more difficult, slower and increases the computational com-

plexity. In contrast, WaveQ exploits the conventional stochastic gradient descent method while jointly optimizing for the original training loss while softly constraining it to simultaneously *learn* the quantized parameters, and more importantly bitwidths. The differentiability of the adaptive sinusoidal regularizer enables simultaneously learning both the bitwidths and pushing the weight values to the quantization levels. As such, WaveQ can be used as a complementary method to some of these efforts, which is demonstrated by experiments with both DoReFa-Net (Zhou et al., 2016) and WRPN (Mishra et al., 2018). Our preliminary efforts (Anonymous) and another concurrent work (Naumov et al., 2018) use a sinusoidal regularization to push the weights closer to the quantization levels. However, neither of these two works make the period a differentiable parameter nor find bitwidths during training.

**Quantized training algorithms** There have been several techniques (Zhou et al., 2016; Zhu et al., 2017; Mishra et al., 2018) that train a neural network in a quantized domain after the bitwidth of the layers is determined manually. DoReFa-Net (Zhou et al., 2016) uses straight through estimator (Bengio et al., 2013) for quantization and extends it for any arbitrary $k$ bit quantization in weights, activations, and gradients. WRPN (Mishra et al., 2018) is training algorithm that compensates for the reduced precision by increasing the number of filter maps in a layer (doubling or tripling). TTQ (Zhu et al., 2017) quantizes the weights to ternary values by using per layer scaling coefficients learnt during training. These scaling coefficients are used to scale the weights during inference. PACT (Choi et al., 2018a) proposes a technique for quantizing activations by introducing an activation clipping parameter $\alpha$. This parameter ($\alpha$) is used to represent the clipping level in the activation function and is learned via back-propagation during training. More recently, VNQ (Achterhold et al., 2018) uses a variational Bayesian approach for quantizing neural network weights during training. VNQ requires a careful choice prior distribution of the weights, which is not straightforward, and the model is often intractable. In contrast, WaveQ is directly applicable without introducing extra hyperparameters to optimize. Additionally, VNQ takes on a probabilistic approach, while WaveQ is a deterministic approach towards soft quantization.

**Loss-aware weight quantization.** Recent works pursued loss-aware minimization approaches for quantization. (Hou et al., 2017) and (Hou & Kwok, 2018) developed approximate solutions using proximal Newton algorithm to minimize the loss function directly under the constraints of low bitwidth weights. One effort (Choi et al., 2018b) proposed to learn the quantization of DNNs through a regularization term of the mean-squared-quantization error. LQ-Net (Zhang et al., 2018) proposes to jointly train the network and its quantizers. DSQ (Gong et al., 2019) employs a series of tanh functions to gradually approximate the staircase function for low-bit quantization (e.g., sign for 1-bit case), and meanwhile keeps the smoothness for easy gradient calculation. Although some of these techniques use regularization to guide the process of quantized training, none explores the use of adaptive sinusoidal regularizers for quantization. Most recently, (Nguyen et al., 2020) suggests using $|cos|$ function as a regularizer. Moreover, unlike WaveQ, these techniques do not find the bitwidth for quantizing the layers.

**Techniques for discovering quantization bitwidths.** A recent line of research focused on methods which can also find the optimal quantization parameters, e.g., the bitwidth, the stepsize, in parallel to the network weights. Recent work (Ye et al., 2018) based on ADMM (adm) runs a binary search to minimize the total square quantization error in order to decide the quantization levels for the layers. Most recently, (Uhlich et al., 2019) proposed to indirectly learn quantizer's parameters via Straight Through Estimator (STE) (Bengio et al., 2013) based approach. In a similar vein, (Esser et al., 2019) has proposed to learn the quantization mapping for each layer in a deep network by approximating the gradient to the quantizer step size that is sensitive to quantized state transitions. On another side, recent works (Elthakeb et al., 2018; Wang et al., November 21, 2018) proposed a reinforcement learning based approach to find an optimal bitwidth assignment policy.

**Quantizing Transformers.** FullyQT (Prato et al., 2019) uses a bucketing based uniform quantization proposed by QSGD (Alistarh et al., 2016) and extends it to Tranformers. Q8BERT (Zafrir et al., 2019) quantizes all the GEMM (General Matrix Multiply) operations to 8 bit by adding an additional term for quantization loss during training, which is calculated based on the rounding effect of floating point values (Shaw et al., 2018). WaveQ, however, uses a sinusoidal regularizer to automatically push the weights towards the quantization levels.

## 7 CONCLUSION

This paper devised WaveQ that casts the two problems of finding layer bitwidth and quantized weights as a gradient-based optimization through parametric sinusoidal regularization. WaveQ provides significant improvements over the state-of-the-art and is even applicable to the Transformers.

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

# Supplementary Material for "WaveQ: Gradient-Based Deep Quantization of Neural Networks through Sinusoidal Regularization"

## A   DETAILED THEORETICAL ANALYSIS

### A.1   MOTIVATION

The results of this section are motivated by the following question.

**Question A.1.** *Suppose that a function $F : \mathbb{R}^n \to [0, \infty)$ has many global minima and that $Q \subset \mathbb{R}^n$ is closed. How do we isolate the global minima of $F$ that are closest to $Q$ without actually computing the full set of global minima of $F$?*

Intuitively, we would like to show that if $\epsilon > 0$ is very small, then the global minima of the function

$$F(x) + \epsilon d(x, Q)$$

are very close to the global minima of $F$ closest to $Q$. To achieve this we will have to introduce first the concept of convergence of sets and then we will show that our intuition is correct by proving that the set of global minima to the above relaxed function converges to a subset of global minima of $F$ closest to $Q$.

### A.2   RELEVANT DEFINITIONS

**Definition A.2.** If $F : \mathbb{R}^n \to [0, \infty)$ satisfies $\lim_{|x| \to \infty} F(x) = +\infty$, we will say that $F$ is coercive.

**Definition A.3.** For a coercive function $F : \mathbb{R}^n \to [0, \infty)$ we let $S_F = \{x \in \mathbb{R}^n : F(x) = \min_{y \in \mathbb{R}^n} F(y)\}$ be coercive.

**Lemma A.4.** *Assume that $F : \mathbb{R}^n \to [0, \infty)$ is continuous and coercive. Then $F$ has at least one global minimum. That is, $S_F$ is non-empty. Furthermore, $S_F$ is a compact set.*

**Definition A.5.** Let $F, G : \mathbb{R}^n \to [0, \infty)$ are continuous and assume that $F$ is coercive. Define

$$S_{F,G} = \{x \in S_F : G(x) = \inf_{y \in S_F} G(y)\},$$

the minima of $F$ which minimize $G$ among the minima of $F$.

**Definition A.6.** Let $Q \subset \mathbb{R}^n$ be a closed set and assume that $x \in \mathbb{R}^n$. Define the distance from $x$ to the set $Q$ to be

$$d(x, Q) = \inf_{y \in Q} \|x - y\|.$$

Observe that since $Q$ is a closed set we have that $x \in Q$ if and only if $d(x, Q) = 0$ and otherwise $d(x, Q) > 0$.

**Definition A.7.** Let $A, B \subset \mathbb{R}^n$ be compact sets. We define the Hausdorff distance between $A$ and $B$ by

$$d_H(A, B) = \max\{\sup_{x \in B} d(x, A), \sup_{y \in A} d(y, B)\}.$$

Observe that $d_H(A, B) = 0$ if and only if $A = B$.

**Definition A.8.** Let $\{S_\delta\}_{\delta > 0}$ be a family of compact subsets of $\mathbb{R}^n$. We say that $\lim_{\delta \to 0} S_\delta = S_*$ if

$$\lim_{\delta \to 0} d_H(S_\delta, S_*) = 0.$$

**Lemma A.9.** *Let $S_\delta$ be a family of compact subsets of $\mathbb{R}^n$, then $\lim_{\delta \to 0} S_\delta = S_*$ if and only if the following two conditions hold.*

    *1. If $x_\delta \in S_\delta$ converges to $x$, then $x \in S_*$*

    *2. For every $x \in S_*$, there exists a family $x_\delta \in S_\delta$ with $x_\delta \to x$.*

The lemma is just an exercise in the definition.

## A.3 STATEMENT OF THE THEOREM

**Theorem 2.** *Let $F, G : \mathbb{R}^n \to [0, \infty)$ are continuous and assume that $F$ is coercive. Consider the sets $S_{F+\delta G}$, the set of points at which $F + \delta G$ is globally minimum. The following are true:*

1. *If $\delta_n \to 0$ and $S_{F+\delta_n G} \to S_*$, then*

$$S_* \subset S_{F,G}$$

2. *If $\delta_n \to 0$ then there is a subsequence $\delta_{n_k} \to 0$ and a non-empty set $S_* \subset S_{F,G}$ so that $S_{F+\delta_{n_k}G} \to S_*$.*

*Proof.* The second statement follows from the standard theory of Hausdorff distance on compact metric spaces and the first statement. For the first statement, assume that $S_{F+\delta_n G} \to S_*$. We wish to show that $S_* \subset S_{F,G}$. Assume that $x_n$ is a sequence of global minima of $F + \delta_n G$ converging to $x_*$. It suffices to show that $x_* \in S_{F,G}$. First let us observe that $x_* \in S_F$. Indeed, let

$$\lambda = \inf_{x \in \mathbb{R}^n} F(x)$$

and assume that $x \in S_F$. Then,

$$\lambda \le F(x_n) \le (F + \delta_n G)(x_n) \le (F + \delta_n G)(x) = \lambda + \delta_n G(x) \to \lambda.$$

Thus, since $F$ is continuous and $x_n \to x_*$ we have that $F(x_*) = \lambda$ which implies $x_* \in S_F$. Next, define

$$\mu = \inf_{x \in S_F} G(x).$$

Let $\hat{x} \in S_{F,G}$ so that $G(\hat{x}) = \mu$. Now observe that, by the minimality of $x_n$ we have that

$$\lambda + \delta_n \mu = (F + \delta_n G)(\hat{x}) \ge (F + \delta_n G)(x_n) \ge \lambda + \delta_n G(x_n)$$

Thus,

$$G(x_n) \le \mu$$

for all $n$. Since $G$ is continuous and $x_n \to x_*$ we have that $G(x_*) \le \mu$ which implies that $G(x_*) = \mu$ since $x_* \in S_F$. Thus, $x_* \in S_{F,G}$. $\square$

## B QUANTIZER

Here, we give an overview about the used quantization method. Consider a floating-point variable $w_f$ to be mapped into a quantized domain using $(b + 1)$ bits. Let $\mathcal{Q}$ be a set of $(2k + 1)$ quantized values, where $k = 2^b - 1$. Considering linear quantization, $\mathcal{Q}$ can be represented as $\{-1, -\frac{k-1}{k}, ..., -\frac{1}{k}, 0, \frac{1}{k}, ..., \frac{k-1}{k}, 1\}$, where $\frac{1}{k}$ is the size of the quantization bin. Now, $w_f$ can be mapped to the $b$-bit quantization (Zhou et al., 2016) space as follows:

$$w_{qo} = 2 \times \text{quantize}_b \left( \frac{\tanh(w_f)}{2 \max(|\tanh(W_f)|)} + \frac{1}{2} \right) - 1 \tag{B.1}$$

where $\text{quantize}_b(x) = \frac{1}{2^b - 1} \text{round}((2^b - 1)x)$, $w_f$ is a scalar, $W_f$ is a vector, and $w_{qo}$ is a scalar and *tanh* is used to limit its range to $[-1, 1]$. Then, a scaling factor $c$ is determined per layer to map the final quantized weight $w_q$ into the range $[-c, +c]$. As such, $w_q$ takes the form $cw_{qo}$, where $c > 0$, and $w_{qo} \in \mathcal{Q}$.

These learned parameters $(b, \alpha)$, as explained in Section 2.2, can be mapped to the quantizer parameters explained in Equation equation B.1. For $(b + 1)$ bits quantization (the extra bit is the sign bit):

$$k = 2^b - 1, \quad \text{and} \quad c = \alpha = 2^b / 2^\beta \tag{B.2}$$

Table 4: Hyperparameters settings.

| Network | # Epochs | | | Batch Size | Learning Rate | Weight Decay | Momentum |
|---|---|---|---|---|---|---|---|
| | Phase 1 | Phase 2 | Phase 3 | | | | |
| SimpleNet on CIFAR10 | 59 | 74 | 44 | 128 | 0.001 | 0.0001 | 0.9 |
| VGG11 on CIFAR10 | 83 | 104 | 62 | 128 | 0.0001 | 0.0001 | 0.9 |
| SVHN-8 on SVHN | 36 | 45 | 27 | 128 | 0.01 | 0.0001 | 0.9 |
| ResNet-20 on CIFAR10 | 32 | 41 | 24 | 128 | 0.01 | 0.0001 | 0.9 |
| AlexNet on ImageNet | 28 | 35 | 21 | 256 | 0.01 | 0.0001 | 0.9 |
| ResNet-18 on ImageNet | 32 | 40 | 24 | 256 | 0.01 | 0.0001 | 0.9 |
| MobileNet on ImageNet | 41 | 52 | 31 | 256 | 0.01 | 0.0001 | 0.9 |

## C CONVERGENCE ANALYSIS

Figure 8 (a), (b) show the convergence behavior of WaveQ by visualizing both accuracy and regularization loss over finetuning epochs for two networks: CIFAR10 and SVHN. As can be seen, the regularization loss (WaveQ Loss) is minimized across the finetuning epochs while the accuracy is maximized. This demonstrates a validity for the proposed regularization being able to optimize the two objectives simultaneously. Figure 8 (c), (d) contrasts the convergence behavior with and without WaveQ for the case of training from scratch for VGG-11. As can be seen, at the onset of training, the accuracy in the presence of WaveQ is behind that without WaveQ. This can be explained as a result of optimizing for an extra objective in case of with WaveQ as compared to without. Shortly thereafter, the regularization effect kicks in and eventually achieves $\sim 6\%$ accuracy improvement.

The convergence behavior, however, is primarily controlled by the regularization strengths $(\lambda_w, \lambda_\beta)$. As briefly mentioned in Section 2.2, $(\lambda_w, \lambda_\beta) \in [0, \infty)$ is a hyperparameter that weights the relative contribution of the proposed regularization objective to the standard accuracy objective.

We reckon that careful setting of $\lambda_w, \lambda_\beta$ across the layers and during the training epochs is essential for optimum results (Choi et al., 2018b).

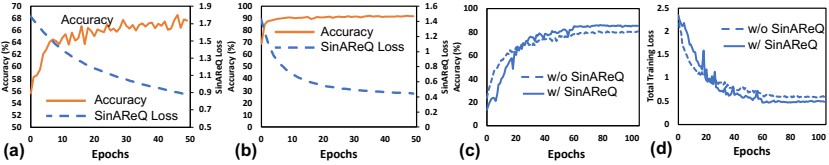

Figure 8: Convergence behavior: accuracy and WaveQ regularization loss over fine-tuning epochs for (a) CIFAR10, (b) SVHN. Comparing convergence behavior with and without WaveQ during training from scratch (c) accuracy, (d) training loss. Network: VGG-11, 2-bit DoReFa quantization

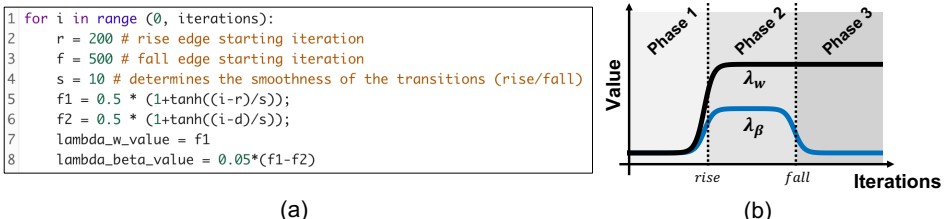

```
1  for i in range (0, iterations):
2      r = 200 # rise edge starting iteration
3      f = 500 # fall edge starting iteration
4      s = 10 # determines the smoothness of the transitions (rise/fall)
5      f1 = 0.5 * (1+tanh((i-r)/s));
6      f2 = 0.5 * (1+tanh((i-d)/s));
7      lambda_w_value = f1
8      lambda_beta_value = 0.05*(f1-f2)
```

(a)                                                          (b)

Figure 9: Math formula for setting $\lambda_w$ and $\lambda_\beta$ during training iterations.

Table 5: Performance of WaveQ on BERT.

| MODEL: CamemBERT BITWIDTH: W4,5/A8 | SPOKEN | | PARTUT | |
|---|---|---|---|---|
| | UPOS | LAS | UPOS | LAS |
| Baseline (FP) | 96.99 | 81.37 | 97.65 | 93.43 |
| Quantized w/ Unregularized Finetuning | 89.91 | 72.32 | 90.89 | 84.25 |
| **Quantized w/ WaveQ Regularized Finetuning** | **93.41** | **79.34** | **95.76** | **91.55** |

Table 6: Validation top-1 accuracy for training from scratch w/ WaveQ vs w/o WaveQ.

| | CIFAR10 (FP Accuracy = 74.53 %) | | | SVHN (FP Accuracy = 96.4 %) | | |
|---|---|---|---|---|---|---|
| | 3 bits | 4 bits | 5 bits | 3 bits | 4 bits | 5 bits |
| Training w/o WaveQ | 9.6 | 31.8 | 70.3 | 61.7 | 79.1 | 90.6 |
| Training w/ WaveQ | 44.8 | 66.6 | 73.2 | 79.3 | 85.1 | 94.8 |
| Improvement (%) | (+35.2)⬆ | (+34.8)⬆ | (+2.9)⬆ | (+17.6)⬆ | (+6.0)⬆ | (+4.2)⬆ |

## D   WAVEQ PERFORMANCE ON BERT

Additionally, Table 5 provides layer-wise quantization with a heterogeneous mix of 4 and 5 bits for the BERT model. In all cases, WaveQ improves UPOS and LAS metrics for two French treebanks (SPOKEN, PARTUT).

## E   TRAINING FROM SCRATCH

Table 6 shows a comparison between training from scratch with WaveQ vs without. It can be seen that incorporating WaveQ into the training process achieves strictly better accuracy than the baseline training without WaveQ across all cases. Moreover, higher improvements are obtained at lower bitwidths reaching to $35\%$

## F   REGULARIZATION STRENGTHS

Having a regularization strength is a normal setting associated with any regularization method. The criterion for choosing $\lambda_w$ and $\lambda_\beta$ is to balance the magnitude of regularization loss to be smaller than the magnitude of accuracy loss. We then perform a grid search over a few points and chose the ones with the best convergence.

From the theoretical perspective, while the theorem is stated in terms of a limit as the regularization parameter vanishes, the proof in fact gives a corresponding stability result. Namely, if the regularization parameter is sufficiently small relative to the main loss then the minimizers will be "almost" quantized.

## G   HETEROGENEOUS COMPARISON

For Mobilenet-V2, WaveQ quantizes the network to an average bitwidth of 3.95(20.66 GBOPS) compared to 5.90(29.16 GBOPs) reported by Uhlich et al. (2019). Similarly, for Resnet-18, WaveQ achieves an average bitwidth of 3.57(62.56 GBOPs) compared to 5.47(65.90 GBOPs) by Uhlich et al. (2019). Table 7 shows this comparison. However this is not a fair comparison since WaveQ is a regularization method and not a full-blown quantization technique.

Table 7: Comparison to heuristic-based bitwidth selection method.

| | MobileNet-V2 | | | ResNet-18 | | |
|---|---|---|---|---|---|---|
| | Accuracy (Learned/FP Baseline) | GBOPs | Avg. BW | Accuracy (Learned/FP Baseline) | GBOPs | Avg. BW |
| Ref [31] (Learned) | 70.5%/70.2% | 2002.30 | 5.90 | 70.6%/70.3% | 414.42 | 5.47 |
| **Ours (Learned)** | **65.8%/71.8%** | **1418.26** | **3.95** | **70.0%/70.1%** | **392.76** | **3.57** |

