# OpenReview forum: "WAVEQ: GRADIENT-BASED DEEP QUANTIZATION OF NEURAL NETWORKS THROUGH SINUSOIDAL REGULARIZATION"
_ICLR.cc/2021/Conference — Reject_

### Official Review · AnonReviewer2 · 2020-10-25
**Interesting Motivation but Unfortunately  Unsatisfactory Empirical and Theoretical Results**

**Rating:** 4
**Confidence:** 5

**Review:**

The paper proposes WaveQ which proposes a sinusoidal regularization approach for quantizing Neural Networks. The motivation is to enable mixed-precision quantization, since homogeneously quantizing the model to low precision can lead to accuracy degradation.

The main problem with heterogeneous/mixed-precision bit-width is that its search space is exponentially large. To address this the authors propose to automatically learn this bit-precision by adding a sinusoidal regularization where the period learn able.

While the approach is interesting but the theoretical and empirical results are not satisfactory and as such it is not clear how the proposed method is superior to other methods proposed in the literature. In particular:


- The theoretical analysis provided does not apply to the proposed method. This is because the theory requires that the regularization function vanish after many iterations, whereas in the experiments the opposite approach is used.

- The accuracy results provided in the empirical section incur significant degradation as compared to the baseline. Furthermore, the comparison is performed with old quantization methods. Newer mixed-precision quantization results using for example HAQ leads to much better quantization. As such, it is not clear what is the advantage of the proposed method?

- How is this approach different than Achterhold et al., 2018?

---

> ### Author Response · Authors · 2020-11-24
> **Author Response**
>
>
> Thank you for the insightful and stimulating comments.
>
> ===== (1) The theoretical analysis ===========
>
> It seems that there is a misunderstanding in interpreting our theoretical and experimental results, accordingly we provide more details to clarify this.
>
> First, we would like to clarify that while the theorem is stated in terms of a limit as the regularization parameter vanishes, the proof in fact gives a corresponding stability result. Namely, if the regularization parameter is sufficiently small relative to the main loss then the minimizers will be "almost" quantized.
>
> Second, we provide results to the updated paper comparing two profiles of the regularization strength ($\lambda_w$).
>
> -- Profile 1: $\lambda_w$ gradually increases as training proceeds then gradually decays towards the end of training. (Figure 7(a)).
>
> -- Profile 2: $\lambda_w$ gradually increases as training proceeds and remains high (Figure 7 (c)).
>
> Figure 7 (a,c) depicts different loss components and Figure 7 (b,d) visualizes weights trajectories.
> Both profiles show that accuracy loss is unimpededly minimized along with WaveQ loss.
>
> Our theoretical results (as originally stated) align with Profile 1 (Figure 7(b)) . With Profile 2, although $\lambda_w$ decays back towards the end of training, the weights mostly remain tied to their quantization levels except for a few deflections that cause slight increase of the regularization loss towards end of training. In terms of test accuracy, both profiles yield similar results (Profile 1, $74.95$%) vs (Profile 2, $74.45$%).
>
> ===== (2) Baseline improvement and comparison to HAQ ========
>
> Baseline improvements:
>
> Our results (as reported in Table 1) show that the incorporation of WaveQ consistently enhances the performance of existing techniques and outperforms multiple state-of-the-art methods considered as our baseline.
>
> WaveQ in comparison to HAQ:
>
> HAQ only provides a method for bitwidth selection using reinforcement learning. Such methods have a high computational complexity as the bitwidth policy must be learned, which involves many iterations of finetuning/evaluation for different bitwidth combinations for the DNN under consideration.
>
> In terms of runtime comparison, reinforcement learning based methods, such as HAQ, typically take up to multiple days to find optimum combination of quantization bitwidths for a single network, while gradient based methods such as WaveQ can finish in hours as it involves a single training pass while biggybagging on stochastic gradient descent without extra computation cost.
>
> In terms of scalability, such high computational cost, in addition to being sample inefficient, prohibits reinforcement learning based methods, like HAQ, from being scalable to very deep networks. In contrast, WaveQ does not incur scalability overhead, so it can seamlessly extend to very deep networks.
>
> Lastly, we would like to emphasize that WaveQ is a quantization-aware regularization method rather than being an explicit quantized training method. In this sense, WaveQ is a complementary approach as it can be applied on top of other existing quantization methods, such as HAQ, DoReFa, etc. Incorporating WaveQ during conventional training is orthogonal to other methods and provides a better starting point for any post-training based quantization methods.
>
> ==== (3) Comparison to Achterhold et al., 2018 =====
>
> The following highlight some of the differences.
>
> (1) WaveQ also offers to learn the bitwidth per-layer granularity; while VNQ does not, instead it chooses the numbers of bits of precision manually.
>
> (2) VNQ takes a variational approach and requires a careful choice prior distribution of the weights, which is not straightforward, and the model is often intractable. In contrast, WaveQ is directly applicable without introducing extra hyperparameters to optimize.
>
> (3) VNQ involves approximations, for example a closed-form expression for the KL divergence cannot be obtained, so they use a differentiable approximation. WaveQ involves no approximations by employing unconditionally continuous and differentiable objectives.
>
> (4) VNQ takes on a probabilistic approach, while WaveQ is a deterministic approach towards soft quantization.
>
> (5) VNQ provides results for ternary quantization only (bitwidth = 2). In contrast, WaveQ directly supports arbitrary bitwidth quantization and provides results for a variety of bitwidths.

---

### Official Review · AnonReviewer4 · 2020-10-26
**Good paper, suggests a useful method for quantization-aware training**

**Rating:** 7
**Confidence:** 4

**Review:**

The paper proposes using a sinusoidal regularizer for neural network quantization. The regularizer “WaveQ” (sin^2) pushes floating-point parameters towards quantized values. Because the period of the function is highly related to the required bit-width, it can be used to determine the bit-width while keeping good characteristics - continuous and trainable. The authors provide experiments on both CNN and Transformers. The proposed method is widely adaptable and easy-to-use with quite promising results.

Major questions/suggestions:

1.	In equation 2.1, the period (=quantization step) is s= 1/ (2^beta – 1), so the regularizer tries to move ‘w’ towards the multiple of ‘s’. As I understand, the desired range of ‘w’ is (-1, 1), to properly match the actual quantized value. But there is no clipping nor normalization applied to ‘w’. Is there something I missed?

2.	There is a missing work [Nyuyen, 2020] which suggests a similar regularizer (|cos|). However, this paper is still valuable and does not hurt novelty.
[Ngugen, 2020] Quantization Aware Training with Absolute-Cosine Regularization for Automatic Speech Recognition

3.	It seems that WaveQ is also used for activation quantization (Section 4.1.), but there are not enough details about applying WaveQ for activations.

Minor issues:

1.	Equation B.2 is frequently used in Section 2. Also, the ‘alpha’ term needs more explanation when it first appears. (‘scaling factor’ is not enough) Consider moving the equation B.1, B.2. to the front.

2.	Many recent quantization papers use w / max(|w|) to normalize weight. Is there any advantage to use tanh(w) / max(tanh(w)) in Equation B.1?

3.	Typo: ‘bitwidh’ in Sec.4.1, ‘citep’ in Sec.5, ‘ofcitepp’ in Sec.6,

4.	I believe the theoretic part is OK but am not sure about the exactness and its importance.

---

> ### Author Response · Authors · 2020-11-24
> **Author Response**
>
> Thank you for the insightful and encouraging comments.
>
> ======= (1) Clipping and the use of tanh (major.1 and minor.2 ) ==========
>
> Thanks for the detailed review and the observation. In fact, tanh is used to limit the value range to [-1,1], so smooth clipping is implied in Equation B.1. Accordingly, we clarified that further in the updated text.
>
>
> ======= (2) Updating missing related work (major.2) ==========
>
> Thanks for pointing that out, we have updated our related work to include it.
>
>
> ======= (3) Activation quantization (major.3) ==========
>
> In the reported experiments, the proposed regularization method only applied to weights. For activation quantization, we utilized an existing quantization method (DoReFa), and compared the end-to-end results with vs without the proposed regularization.
>
>
> ======= (4)  Moving equation B.1, B.2 to the front (minor.1) ==========
>
> Thank you for the insightful suggestion, accordingly we moved the Quantizer subsection (including equations B.1, B.2) to the main paper in Section 2.1.
>
>
> ======= (5) Typos (minor.3) ==========
>
> Thanks, fixed in the updated version.

---

### Official Review · AnonReviewer1 · 2020-10-28
**Borderline**

**Rating:** 5
**Confidence:** 4

**Review:**

This paper proposed a regularization term to control the bit-width and encourage the DNN weights moving to the quantization intervals. The key in such regularization is the Sinusoidal function, where the penalty is maximized in the middle of quantization levels and minimized at the quantization points. The sinusoidal period is regarded as the continuous representation of the bit-width.

Pros:
1. The paper is well written and easy to understand. Illustration figures help a lot to present the proposed method. E.g., Figure 2 clearly shows the key idea of this paper.

2. The idea of using the sinusoidal period as a continuous representation is novel and it makes it natural to use gradient descent methods to optimize the bit-width.

3. Experiments on different datasets and tasks show that the proposed method can bring improvement on accuracy by using better regularization and / or better bit-width allocation.

Cons:
1. The quantization regularization can only be used on weight quantization, while activation quantization is very important and sometimes more sensitive to the accuracy drop. On this point, the proposed method cannot be used to allocate the bit-width of activation, while activations can actually have different bit-width to the weights.

2. Some related works are missing.
For quantization regularization, it is used in ProxQuant[1] as well, where the weights are regularized with a norm-based quantization regularization. It also discussed the similar observation that the regularization can help quantization-aware training.
For bit-width allocation, [2] formulated the bit-width allocation as a knapsack problem and used ADMM to iteratively optimize the NN compression. [3] used Bayesian optimization to allocate the bit-width for different layers.

3. The procedure to tune the regularization strengths is a little complicated. Although the authors proposed a detailed procedure for tuning these hyper-parameters, it's questionable that the same procedure can be widely applied in different networks/optimizers/datasets. For $\lambda_\beta$, it actually determines the number of total bits for the entire network, so it should depends on the expected compression rate of the quantized DNN.

4. Using the sinusoidal period to decide the bit-width has an assumption that the range of the weights is fixed: otherwise one can reduce both the period and value range to keep the bit-width unchanged. However, the range of weights is usually not fixed in the training.

5. The experiments on bit-width allocation may be not enough to show the effectiveness of the proposed method as a bit-width allocation method. It's better to compare with some other bit-width allocation methods.

In general, my rating is borderline. I hope the authors can give some response to the cons listed above.

Reference:

[1] Bai, Y., Wang, Y.X. and Liberty, E., 2018. Proxquant: Quantized neural networks via proximal operators. arXiv preprint arXiv:1810.00861.

[2] Yang, H., Gui, S., Zhu, Y. and Liu, J., 2020. Automatic Neural Network Compression by Sparsity-Quantization Joint Learning: A Constrained Optimization-Based Approach. In CVPR 2020.

[3] Tung, F. and Mori, G., 2018. Clip-q: Deep network compression learning by in-parallel pruning-quantization. In CVPR 2018.

---

> ### Author Response · Authors · 2020-11-24
> **Author Response**
>
> Thank you for the insightful and stimulating comments.
>
> === (1) Activation quantization ===
>
> This paper focuses on weight quantization and simultaneously learning the bitwidths, which is unprecedented. Due to this unique and novel gradient-based approach in quantization, it can be applied on top of other activation quantization techniques and due to its gradient-based nature even complement them better since WaveQ adjusts the weights while considering the effects of activation quantization in its updates. WaveQ actually has been used in consort with activation quantization for results in Table 1.
>
> Moreover, weight quantization has its own significant merit and benefits. For instance, weight quantization alone provides orders of magnitude energy reduction as energy consumption is dominated by memory access (for example, DRAM memory access energy is around 3 orders of magnitude of an add operation).
>
> === (2) Comparison to other related works ===
>
> Thanks for the pointer to these related work, which is included in the updated version.  The following highlights some fundamental differences.
>
> (1)  WaveQ not only regulates weights towards quantized values, but also learns the bitwidth of the layers by making the period of the sinusoidal regularizer a trainable parameter. While none of these works learn the bitwidth.
>
> (2) All the related methods define a new optimization problem and use a special method for solving it. For example, [1] uses a proximal gradient method (adds a prox step after each stochastic gradient step), [2] uses ADMM, and [3] takes a Bayesian approach. This only makes the training more difficult, slower and increases the computational complexity. In contrast, WaveQ exploits the conventional stochastic gradient descent method while jointly optimizing for the original training loss while softly constraining it to simultaneously learn the quantized parameters, and more importantly bitwidths.
>
> (3) WaveQ incorporates an unconditionally continuous and differentiable objective with arbitrary bitwidth and that is how it seamlessly blends with the conventional training loss in a unified optimization setting. Other works get around the inherent discontinuities in the quantization process by incorporating various approximations. For example, the resulting regularizer in [1] is a W-shaped non-smooth regularizer for binary quantization.
>
>
> === (3) Regularization strengths ===
>
> (1) Hyperparameters setting (such as learning rates and regularization strengths) is an imperative part of the general development of deep neural networks more than being specific to the proposed method. And each quantization technique has its own hyperparamters.
>
> (2) The particular procedure followed and illustrated in the paper is intuitive and its complexity is comparable to other procedures of similar studies. Further, several experiments, as reported in Tables I, II, and III, established the applicability of the used procedure across 8 different neural networks (one of them is Transformer model), with 4 different datasets.
>
> (3) The bitwidth regularization strength $\lambda_\beta$ is multiplied by the continuous variable $\beta$ which is proportional to the compression rate; that is the sinusoidal period. That entire product forms the second objective which is what goes into the optimizer. The sole purpose of $\lambda_\beta$ is to define an intermediate window for learning the bitwidth and consequently the compression rate while balancing that relative to other objectives.
>
> === (4) min-max ranges  ===
>
> (1) Determining the min-max range of the weights is an issue associated with quantization-aware training methods other than specific to this proposed method, and WaveQ is no different than other techniques
>
> (2) For finetuning purposes, the common approach is statically quantizing the weights to a single min-max range, since weights have already occupied a stable range which does not tend to change much during finetuning.
>
> (3) For training from scratch, some adaptive techniques could be incorporated. In our case, however, during phase one (Figure 2 (e) ), we let the network learn the range of the weights considering the original task loss only, while both $\lambda_\beta$ and $\lambda_w$ are set to zeros. By the time we transition to phase 2, the larger $\lambda$’s gradually engage the regularization, we adopt the min-max range as learnt during phase 1.
>
> === (5) Comparison to other bit-width allocation methods  ===
>
> We added one more comparison against [1] to the updated paper. Table 7 shows this comparison results. In summary, for MobileNet-V2, WaveQ quantizes the network to an average bitwidth of 3.95 (that is 20.66 Giga Bits per OPerations  (GBOPs)) compared to 5.90 (that is 29.16 GBOPs) reported by [1]. Additionally, for ResNet-18, WaveQ achieves an average bitwidth of 3.57 (62.56 GBOPs) compared to 5.47 (65.90 GBOPs) by [1].
>
> [1] Uhlich, et al., Mixed precision DNNs: All you need is a good parametrization. ICLR20

---

### Official Review · AnonReviewer3 · 2020-10-29
**How to include bitwidths in your training objective**

**Rating:** 7
**Confidence:** 4

**Review:**

When training quantized neural networks, one typically first fixes the desired bitwidth $b$ (of weights and activations). While training, one maintains and updates full-bitwidth weights during backprop and "cheats" by using $b$-quantized versions of these full-precision weights during forward propagation. The quantization scheme used may vary.

What if we wished instead to *learn* $b$? This is especially attractive, for instance, if we wish a distinct $b_i$ for each layer $i$. We could introduce a variable $\beta_i$ to represent $b_i$ during training. We would probably add a regularizer to minimize $\Sigma \beta_i$ to encourage training toward small bitwidths. However, this still leaves the problem that the $\beta_i$ would be real, not integer, as required for quantization. We could cheat here by using $\lceil \beta_i \rceil$, but note that quantization error $\beta_i - \lceil \beta_i \rceil$ may be quite large relative to the small integer values $\beta$ is expected to take. Is there a way to ensure that training automatically produces $\beta$s that are close to integer values?

This is where the current paper introduces a neat trick. By embedding $\beta$ inside a (suitably scaled) sinusoidal loss function, they ensure that the $\beta$s tend to have integer values (that correspond to the minima of the sinusoid).

I haven't seen this trick before, and it seems like an elegant approach to the problem of favoring integer values within the global optimization process. The goal of producing distinct bitwidths for each layer is definitely a valid one, so this technique does also address a practical problem.

The technique's practical impact seems limited. In particular, it seems that the main effect is to reduce the average bitwidth of SOTA quantized models from 4 to >= 3.6 at roughly the same accuracy. This should correspond roughly to a 10% gain in performance (runtime), which is quite good but not great. I have no idea how to to interpret the power numbers, so I will ignore those for now.

One question I have though is how the technique compares to a decent heuristic variable-bitwidth baseline. The authors do compare to a  "decrement the bitiwidth of a single layer" baseline, which is a good start. However, what if you do something more informed than that as baseline? Can you tell us what bitwidths your techniques chose for Resnet, for instance? Is there a simple pattern there (e.g. high bitwidth for early layers, and lower later on)? Based on this, what is the "best" non-learned bitwidth selection you can come up with?

In any case, the sinusoid trick for learning integer-like values is a good one for people to know, and it seems that the authors have shown that it can be implemented to have practical impact. Even if bitwidth reduction is currently modest, I can imagine follow-up work to increase it. So I support acceptance of the paper.

---

> ### Author Response · Authors · 2020-11-24
> **Author Response**
>
> Thank you for the insightful and encouraging comments.
>
> ===== Comparison to other heuristic methods ======
>
> Regarding comparing to another heuristic variable-bitwidth baseline. We added one more comparison against [1] to the updated paper. Table 7 shows the comparison results. In summary, for MobileNet-V2, WaveQ quantizes the network to an average bitwidth of 3.95 (that is 20.66 GBOPs [2]) compared to 5.90 (that is 29.16 GBOPs) reported by [1]. Additionally, for ResNet-18, WaveQ achieves an average bitwidth of 3.57 (62.56 GBOPs) compared to 5.47 (65.90 GBOPs) also reported by [1].
>
> ===== Rule based bitwidth selection ======
>
> The results show that the bitwidth patterns are complex and very dependent on the specific DNN under-quantization. The following points provide reasoning why the bitwidths do not follow certain patterns.
> (1) Different layers in different DNNs capture different representations with varying degrees of influence on the classification results due to the architectural variations. (2) Because of different degrees of overparameterization, different layers exhibit different levels of tolerance to imprecision. (3) each layer assume a distribution of weights (typically bell-shaped) each of which has a different dynamic range leading to different degrees of robustness to quantization bitwidths; (4) recent experimental work [2] also shows that layers can be categorized as either "ambient" or "critical" towards post-training re-initialization and re-randomization. In other words, no straightforward set of rules is observed to decide bitwidth assignment a priori.
>
> Furthermore, our results (also commensurate with other works in literature) show that, given a particular accuracy-performance tradeoff, there may not be a unique optimum heterogeneous solution, but rather a set of solutions exhibiting different heterogeneities, which further suggests that choosing the optimal bitwidths is not straightforward. Therefore, gradient-based methods that can learn the optimal bitwidth assignment provide a necessary effective step in heterogenous bitwidth quantization.
>
> [1] Uhlich, et al., Mixed precision DNNs: All you need is a good parametrization. ICLR (2020).
>
> [2] Zhang, C., Bengio, S., and Singer, Y. Are all layers created equal? CoRR abs/1902.01996 (2019).

---

### Author Response · Authors · 2020-11-24
**To Reviewers and the Area Chair**

We thank all the reviewers for their encouraging and stimulating comments. We have addressed all the comments and feedback from the reviewers in our revision and provided a detailed answer in the comments section.

In particular, we added the following revisions to the paper.

(1) We updated our theoretical results section (based on comments from AnonReviewer2 regarding the regularization strength profile) clarifying that while the theorem is stated in terms of a limit as the regularization parameter vanishes, the proof in fact gives a corresponding stability result. Namely, if the regularization parameter is sufficiently small relative to the main loss then the minimizers will be "almost" quantized. To illustrate and demonstrate this further, we provided results to the updated paper (Figure 7) comparing two profiles of the regularization strength.

(2) We added Table 7 to the updated version of the paper (in the supplementary material) to provide a comparison against [1] and further demonstrate the effectiveness of the proposed method as a bit-width allocation method following suggestions from AnonReviewer3,2.

(3) We updated the method section by moving the “Quantizer” paragraph into the main paper in Section 2.1, as suggested by AnonReviewer4.

(4) We updated the related work section following suggestions from reviewers AnonReviewer1,2,4.



[1] Uhlich, et al., Mixed precision DNNs: All you need is a good parametrization. ICLR (2020).

---

### Decision · Program_Chairs · 2021-01-07
**Final Decision**

**Decision:**

Reject

**Comment:**

This paper proposed a regularization term to control the bit-width and encourage the DNN weights moving to the quantization intervals. The paper is well-written and the idea of using the sinusoidal period as a continuous representation is novel. However, the theoretical analysis provided are not consistent with the proposed method.

As for the experimental results, the proposed method incurs significant degradation as compared to the baseline, and comparison with recent quantization methods is lacking.